# Short-Term Gaseous Treatments Improve Rachis Browning in Red and White Table Grapes Stored at Low Temperature: Molecular Mechanisms Underlying Its Beneficial Effect

**DOI:** 10.3390/ijms232113304

**Published:** 2022-11-01

**Authors:** Irene Romero, Raquel Rosales, M. Isabel Escribano, Carmen Merodio, M. Teresa Sanchez-Ballesta

**Affiliations:** Department of Characterization, Quality and Safety, Institute of Food Science, Technology and Nutrition (ICTAN-CSIC), Ciudad Universitaria, E-28040 Madrid, Spain

**Keywords:** table grapes, gaseous treatments, low temperature, rachis quality, gene expression

## Abstract

Short-term gaseous treatments improve rachis quality during table grape postharvest, but little is known about the mechanisms involved. In this work, we observed that the application of a 3-day CO_2_ treatment at 0 °C improved rachis browning of Superior Seedless and Red Globe bunches, affecting the non-enzymatic antioxidant system by reducing the total phenolic content, the antioxidant activity and the expression of different *stilbene synthase* genes. Lipid peroxidation levels revealed lower oxidative stress in CO_2_-treated rachis of both cultivars linked to the activation of the enzymatic antioxidant system. Furthermore, whereas a positive correlation was denoted between rachis browning and the accumulation of key ABA regulatory genes in Red Globe bunches, this effect was restricted to *ACS1*, a key synthetic ethylene gene, in Superior Seedless clusters. This work also corroborated the important role of ethylene-responsive factors in the beneficial effect of the gaseous treatment, not only in the berries but also in the rachis. Finally, the application of the gaseous treatment avoided the induction of cell wall-degrading enzyme-related genes in both cultivars, which could favor the maintenance of rachis quality. This work provides new insight into specific responses modulated by the gaseous treatment focused on mitigating rachis browning independently of the cultivar.

## 1. Introduction

Table grapes (*Vitis vinifera* L.) are highly appreciated by consumers due to their excellent organoleptic and nutritional quality, which has contributed to a significant increase in their consumption worldwide in recent years. Table grape clusters are composed of berries and rachis tissues. Berries are classified as non-climacteric fruit and present low physiological activity during postharvest, reaching their optimum acceptability of appearance, flavor and texture while on the vine. The rachis is the vegetative structure that acts as a support for the berries and is the conduction system for nutrients and water, so its deterioration directly affects the quality of the bunch. Moreover, the rachis has an average respiration rate 28 times higher than that of the berries [1].

After harvest, table grape bunches are susceptible to fungal attack and dehydration processes that reduce their quality, observable by weight loss, berry drop and senescence of the rachis. Thus, storage at low temperatures, close to 0 °C, and a relative humidity of 90–95% for 40–100 days, depending on the cultivar, are used for greater efficiency in the control of water loss and the maintenance of quality which help to extend their postharvest shelf life [2]. However, this cold storage is limited due to the susceptibility of table grapes to fungal attacks, caused mainly by *Botrytis cinerea*, and to water loss, which leads to alterations such as rachis browning [2,3]. In this sense, the amount of research devoted to rachis browning is far less than that invested in the main problem: the control of fungal attacks [4]. Moreover, rachis browning influences the appearance of bunches and, although the rachis only represents about 4% of the cluster’s fresh weight, it affects consumer preference, given that a green rachis is automatically associated with freshness. It should be kept in mind that consumer behavior is considered the main cause of food waste in developed countries [5].

In this regard, the application of a high CO_2_ treatment for 3 days at 0 °C maintained berry quality and reduced rachis browning in red (Cardinal and Red Globe), dark (Autumn Royal) and white (Superior Seedless and Dominga) table grapes, compared with bunches stored in the air [6,7,8,9]. A previous work denoted that the short-term gaseous treatment was effective in controlling rachis browning in red Cardinal table grapes, avoiding the activation of ethylene biosynthesis genes and promoting an osmotic adjustment [3]. However, it was not possible to establish a link between abscisic acid (ABA) and rachis deterioration in Cardinal bunches stored at 0 °C [3]. Considering that the key enzymes controlling ethylene and ABA biosynthesis are 1-aminocyclopropane-1-carboxylic acid oxidase (ACO) and 9-cis-epoxycarotenoid dioxygenase (NCED), respectively, Suehiro et al. [10] indicated that ABA and ethylene might contribute to the skin browning of Shine Muscat grapes by activating the expression of *VvNCED1*, *VvACO2* and *VvACO3*. However, in reference to ABA, Cantin et al. [11] reported that its application at veraison improved the rachis quality of Crimson bunches stored at 0 °C. Therefore, it is necessary to deepen the understanding of the role of ABA in postharvest rachis browning and the interaction between ABA and ethylene.

Ethylene response factors (ERFs) are plant-specific transcription factors belonging to the large AP2/ERF multi-gene family, which can bind to the GCC box present in the promoter of pathogenesis-related (PR) genes [12]. In the case of red Cardinal table grapes, the application of a 3-day gaseous treatment activated the transcript accumulation of different *ERFs* in fruit and non-fruit tissues such as skin, pulp, seeds and rachis [13]. To our knowledge, there is only one other study on the application of controlled atmosphere and cytokinin to reduce table grape rachis browning that refers to the expression of an *ERF* [14]. Further investigation is therefore necessary to be able to draw conclusions about its role in table grape postharvest and in particular in rachis browning.

The control of rachis browning by the application of high levels of CO_2_ is known to be related to the induction of the enzymatic antioxidant system [3]. Although the non-enzymatic antioxidant system seems to participate in the response of berries to low temperatures and high levels of CO_2_ [8,15], little is known about its role in rachis browning. Previous works suggested that the phenylpropanoid pathway, which is one of the best-known metabolic pathways in plant cells, appears to be involved in rachis browning [3,16]. Phenolic compounds play an important role in fruit visual appearance. Moreover, monohydroxyphenols and orthodihydroxyphenols are substrates of plant polyphenol oxidase (PPO) enzymes that produce brown polymers, affecting fruit quality [17]. However, so far it is not known whether the application of high levels of CO_2_ could affect the phenol content in the rachis, which plays an important role in the non-enzymatic antioxidant capacity. Moreover, in the phenylpropanoid pathway, the stilbene synthase (STS) uses p-coumaroyl-CoA and three malonyl-CoA molecules to produce stilbenoids [18], including resveratrol and its derivatives, which are classified as phytoalexins. The content of resveratrol can be induced in grape berries in response to biotic and abiotic stresses and by the application of different postharvest treatments [19,20]. Thus, in a previous work, we observed that the accumulation of stilbene compounds through the application of CO_2_ at 0 °C was cultivar-dependent and linked to the expression of different *VviSTSs* in the skin [9]. In relation to the putative role of the stilbenoid pathway in grape browning, Suehiro et al. [21] reported that increases in the expression of a *VvSTS* and in the *trans*-resveratrol content were associated with the skin browning of Shine Muscat grapes. However, there is a lack of information regarding the expression of *STSs* or the accumulation of resveratrol in non-fruit tissues such as the rachis.

The present work proposes a study to elucidate the molecular mechanisms implicated in the maintenance of rachis quality by the application of short-term gaseous treatments in an early-ripening, white table grape cultivar (Superior Seedless, SS) and a mid-ripening, red cultivar (Red Globe, RG). Thus, the objective is to determine the role of the non-enzymatic (total phenolic content, antioxidant capacity and expression of *VviSTSs*) and enzymatic antioxidant (expression of catalase (*VviCAT*) and ascorbate peroxidase (*VviAPX*)) systems in the development of rachis browning in both cultivars. We also want to elucidate the correlation between the maintenance of rachis quality and the modulation of key ethylene (*ACS1* and *ACO1*) and ABA *(VviNCED1* and *VviNCED2*) regulatory genes, as well as different *ERF* genes. Finally, we will analyze whether the differences denoted in rachis senescence between CO_2_-treated and non-treated bunches are related to the expression of genes (*polygalaturonase* (*VviPG*), *expansin* (*VviEXP*), *xylanase* (*VviXYL*), *pectin methylesterase* (*VviPME*) and *cellulase* (*VviCEL*)), which encode different cell wall-degrading enzymes.

## 2. Material and Methods

### 2.1. Plant Material

Table grapes (*Vitis vinifera* L.) from two cultivars, Red Globe and Superior Seedless, were collected in Abarán, Murcia, Spain (latitude: 38°12′00″ N, longitude: 01°24′00″ W, altitude: 173 m) at its optimum state of maturity in July (SS, 14.77 ± 0.75 °Brix) and September (RG, 12.67 ± 0.15 °Brix). Clusters were transported to the ICTAN on the same day of collection, and those that did not present mechanical or pathological defects were randomly divided into two batches and stored at 0 ± 0.5 °C with a relative humidity of 95% in methacrylate booths of 1 m^3^ capacity. Initially, 5 random clusters were separated that constitute the group of “time 0”. Ten boxes with a content of around 3 kg of table grapes per box were stored in each booth. The first batch of bunches was stored under normal atmospheric conditions for a total of 28 days (non-treated fruit). The second batch was stored with a gas mixture containing 20 kPa CO_2_ + 20 kPa O_2_ + 60 kPa N_2_ (CO_2_-treated fruit) for 3 days at 0 °C, and then transferred to the air under the same conditions as the non-treated bunches for 25 days. Five clusters (approximately 300 g per cluster) were sampled periodically (on days 0, 3, 11 and 28), to collect the rachis of each cluster. This material was frozen in liquid nitrogen and stored at −80 °C for further analysis.

### 2.2. Determination of Lipid Peroxidation

Quantification of the end product of lipid peroxidation, malondialdehyde (MDA), was assayed according to Rosales et al. [3] with some modifications. Briefly, 0.1 g of rachis tissue from the different samples was homogenized with 1.5 mL of 5% trichloroacetic acid (TCA) and centrifuged at 10,000× *g* for 15 min. After centrifugation, 250 µL was mixed with 1 mL of 0.5% thiobarbituric acid in 20% TCA. This mixture was incubated at 100 °C for 30 min and then cooled at room temperature. Absorbance was determined at 532 nm and adjusted for non-specific absorbance at 600 nm. Three independent extractions were made for each sample and extracts were analyzed in duplicate. The MDA content was estimated by using a molar extinction coefficient of 155 mmol L^−1^ cm^−1^.

### 2.3. Analysis of Total Phenols Content by Folin–Ciocalteu Method

For the extraction of total phenols, 0.25 g of rachis tissue from CO_2_-treated and non-treated table grapes stored at 0 °C was homogenized with 0.5 mL of a solution of methanol (1% HCl)-acetone (*v/v*) and mixed for 30 min at room temperature. The extracts were centrifuged at 10,000× *g* for 10 min, and the supernatants were collected. These steps were repeated obtaining a final volume of 1 mL. The supernatants were stored at −20 °C. The content of total phenols in the extracts was determined by the Folin–Ciocalteu method [22] and expressed as mg of gallic acid equivalents g^−1^ fresh weight (FW).

### 2.4. Antioxidant Activities Measured via 2,2-Azino-Bis-3-Ethylbenzothiazoline-6-Sulfonic Acid (ABTS) and Ferric Reducing Antioxidant Power (FRAP) Methods

For the determination of the antioxidant activity of RG and SS grapes, the same extracts as for the determination of total phenols were used. ABTS and FRAP methods were performed as referred to by Romero et al. [8].

### 2.5. Relative Gene Expression via Quantitative Real-Time RT-PCR (RT-qPCR)

Total RNA extraction, DNAse treatment and cDNA synthesis were performed according to Romero et al. [13]. Relative gene expression of *VviSTSs*, *VviERFs*, *VviAPX*, *VviCAT*, *VviACS1*, *VviACO1*, *VviNCED1*, *VviNCED2*, *VviPG*, *VviEXP1*, *VviXTH* and *VviPME* were studied in the rachis of non-treated and CO_2_-treated grapes from the two cultivars stored at 0 °C for up to 28 days via RT-qPCR according to Rosales et al. [3]. Gene-specific primers were designed using Primer 3 software [23] and used to amplify specific products (Appendix A). In the case of *VviSTSs*, primers designed by Ciaffi et al. [24] were used. *Actin1* (XM 002282480) from *V. vinifera* was used as the internal control. The specificity of products was validated according to Romero et al. [13].

### 2.6. Statistical Analyses

The software SPSS v. 28.0 (IBM) was used for the statistical analysis. The different data obtained were analyzed through ANOVA (one-way analysis of variance), and their means ± standard deviation were grouped in subsets via the Tukey-b test (*p* < 0.05). The relationship between relative gene expression, total phenolic content and antioxidant activity was described as a Pearson product-moment correlation coefficient (r), *p* < 0.01 or *p* < 0.05.

## 3. Results

### 3.1. Effect of a 3-Day CO_2_ Treatment on Total Phenolic Content and Antioxidant Capacity in the Rachis of SS and RG Table Grapes Stored at 0 °C

The application of a 3-day treatment with high levels of CO_2_ was effective in controlling rachis browning in RG and SS bunches stored for up to 28 days at 0 °C (rachis browning index (RG, 28 d, Air, 3.66 ± 0.47 b; CO_2_, 2.83 ± 0.23 a), (SS, 28 d, Air 3.00 ± 0.40 b; CO_2_, 2.00 ± 0.35 a) [9]; Figure 1). The application of a 3-day gaseous treatment significantly reduced the accumulation of phenolic compounds in RG and SS rachises in comparison with non-treated samples throughout storage at 0 °C (Figure 2). Thus, the total phenolic content did not vary in CO_2_-treated Red Globe samples at 0 °C, with values with no significant differences to those recorded at time 0. However, as early as day 3 of storage until the end, a significant increase was observed in non-treated samples (Figure 2). Likewise, the phenolic content was always significantly higher in non-treated samples of Superior Seedless bunches when compared with CO_2_-treated ones (Figure 2). However, in this case, the total phenolic content increased after 11 days and reached values higher than those observed at time 0.

The antioxidant capacity determined via FRAP and ABTS methods also showed differences between CO_2_-treated and non-treated rachises in both cultivars, and in general, higher values were recorded in non-treated ones. However, while the antioxidant capacity in most of the CO_2_-treated SS samples was similar to that at time 0, the results obtained through both methods in RG bunches showed higher values than those of freshly harvested fruit. Moreover, there was a significant correlation in both cultivars between FRAP/ABTS methods (SS, r = 0.909, *p* < 0.01; RG, r = 0.824, *p* < 0.01) and when analyzing total phenolic content vs. FRAP/ABTS (SS, r = 0.926/r = 0.951, *p* < 0.01; RG, r = 0.692/r = 0.846, *p* < 0.01) (Appendix A).

### 3.2. Effect of a 3-Day CO_2_ Treatment on the Expression of VviSTSs in the Rachis of SS and RG Table Grapes Stored at 0 °C

The search in the NCBI database allowed Ciaffi et al. [24] to identify 31 functional *STS* genes in the *V. vinifera* genome. These authors indicated that it was possible to design specific primers for expression analysis for only 8 out of the 31 genes identified. For the remaining 23 genes, they designed pairs of conserved primers amplifying from two to four very similar sequences. Thus, we have made use of this previously published information and synthetized primers for the expression of *VviSTS7/8*, *VviSTS9-11*, *VviSTS12*, *VviSTS16-18* and *VviSTS27-30* in the rachis of CO_2_-treated and non-treated SS and RG table grapes stored at 0 °C.

The results indicated that the *VviSTSs* analyzed showed a similar pattern of expression independently of the cultivar analyzed (Figure 3). Thus, although the storage of bunches at 0 °C activated the expression of all the *VviSTSs* analyzed in the rachis, it is important to note that this increase was significantly higher in the samples stored in the air.

However, while the differences in RG bunches between the highest accumulations in the non-treated and treated samples were observed as of day 3, these differences were mainly observed on day 28 in SS bunches. In both cultivars, the accumulation of all the *VviSTSs* correlated with each other (*p* < 0.01) (Appendix A).

### 3.3. Effect of a 3-Day CO_2_ Treatment on the Oxidative Stress in the Rachis of SS and RG Table Grapes Stored at 0 °C

In order to corroborate the role of oxidative stress in the deterioration of rachis, the levels of MDA were analyzed in the rachis of CO_2_-treated and non-treated table grapes (SS and RG) stored at 0 °C. The MDA content, in general, was lower in CO_2_-treated samples from both cultivars when compared with non-treated ones (Figure 4). On day 11, the MDA levels of CO_2_-treated samples of Red Globe clusters remained similar to those of freshly harvested bunches, increasing dramatically and peaking in the non-treated samples. Nevertheless, at this point, no differences were observed between treated and non-treated samples of Superior Seedless bunches. At the end of the storage period, the MDA content increased for the first time in CO_2_-treated Red Globe samples, reaching values similar to those of the non-treated rachis. In the case of the Superior Seedless bunches, although MDA levels were higher than in the freshly harvested samples in both treated and non-treated bunches, they were significantly higher in the non-treated ones.

The analysis of the genes codifying antioxidant enzymes such as APX and CAT denoted that the application of high levels of CO_2_ significantly induced the expression of *VviCAT* in both Superior Seedless and Red Globe rachises, just at the end of the gaseous treatment (3 days) (Figure 4). Similar results were observed in the accumulation of *VviAPX*, but only in the SS samples. These increases were transitory in all the cases, decreasing the accumulation to similar (*VviAPX*) or lower levels (*VviCAT*) than those achieved in freshly harvested samples. In the case of non-treated samples, the expression of *VviCAT* only increased on day 3 in RG samples, although the levels were significantly lower than those recorded in CO_2_-treated rachis. Moreover, the expression of *VviCAT* decreased in non-treated samples of both cultivars at the end of the storage period, as was indicated in the case of CO_2_-treated ones. Finally, the accumulation of *VviAPX* showed a different pattern of accumulation. Thus, the expression of *VviAPX* did not vary in non-treated RG samples throughout the storage at 0 °C in comparison with the levels of the freshly harvested rachis. However, the application of the gaseous treatment decreased the accumulation of the transcripts on day 3, and thereafter it reached values similar to samples on day 0 (Figure 4). No significant correlations were found between MDA and the antioxidant genes analyzed in both cultivars (data not shown).

### 3.4. Effect of a 3-Day CO_2_ Treatment on Ethylene and ABA Biosynthesis Gene Expression in the Rachis of SS and RG Table Grapes Stored at 0 °C

In order to corroborate the ethylene involvement in rachis browning, we analyzed changes in the expression pattern of *VviACS1* and *VviACO1* genes that codify the key regulatory enzymes of ethylene biosynthesis in red and white table grape bunches stored at 0 °C (Figure 5).

In SS bunches, results showed an increase in *VviACS1* transcript accumulation both in CO_2_-treated and non-treated samples throughout storage at 0 °C, although there were no significant differences until day 28 when a sharp increase was observed in CO_2_-treated rachis. However, *VviACO1* levels showed a significant decrease from day 11 until the end of the storage period, with higher levels for non-treated samples than for CO_2_-treated rachis (Figure 5). Curiously, the *VviACS1* transcript levels in the RG cultivar showed a significant increase in non-treated samples on day 3 of storage and remained high until day 28, at which point they decreased to levels similar to those at time 0. CO_2_-treated rachis, on the other hand, presented a slight decrease on day 3, which was maintained throughout the storage period. Likewise, *VviACO1* levels showed a distinct pattern of expression compared with SS, with a transient increase on day 3 in non-treated samples and decreasing on day 11 until the end of the storage period to levels below those observed at time 0 (Figure 5). Furthermore, to investigate the role of ABA in the deterioration of rachis from red and white table grapes, we studied the expression profiles of *VviNCED1* and *VviNCED2*. It is important to note that in both cultivars the expression levels of these genes were higher in the non-treated rachis compared with CO_2_-treated ones. A similar trend was observed in their expression levels in SS bunches, where it is interesting to highlight a notable transitory increase in transcript accumulation on day 3 of storage in non-treated rachis, and a significant decrease, compared with time 0, in CO_2_-treated samples (Figure 5). In the RG cultivar, however, the expression of these genes differed from the trend observed in the white cultivar. Thus, while *VviNCED1* presented a double peak increase on days 3 and 28 in non-treated bunches, *VviNCED2* showed a gradual increase in non-treated samples with a peak on day 28 (Figure 5). By contrast, the expression levels of *VviNCED1* and *VviNCED2* decreased or did not vary throughout the storage of CO_2_-treated clusters, compared with time 0.

In terms of correlation, a significant positive correlation was denoted in the Superior Seedless samples between the accumulation of both ABA regulatory genes (r = 0.760, *p* < 0.01), being negative in the case of *VviACS1* and *VviACO1* (r = −0.659, *p* < 0.01). However, the expression of both ethylene and ABA synthesis genes (*ACS1/ACO1* r = 0.530, *p* < 0.05; *NCED1*/*NCED2*, r = 0.580, *p* < 0.01) was significantly correlated in Red Globe bunches (Appendix A). Furthermore, a positive correlation was observed in both cultivars between the expression of *VviACO1* and *VviNCED1* (*p* < 0.05).

### 3.5. Effect of a 3-Day CO_2_ Treatment on VviERFs Expression in the Rachis of SS and RG Table Grapes Stored at 0 °C

With the aim of discerning the role of *ERFs* in the beneficial effect of a 3-day gaseous treatment, five *ERFs* previously studied in red Cardinal table grapes [13] were analyzed.

A significant increase in the accumulation of three of the five *VviERFs* (all except *VviERF10 and VviERF11*) was observed in white SS table grapes at the end of the gaseous treatment (3 days) (Figure 6), being transitory in the case of *VviERF069* and *VviERF6L7*. At this point (day 3), no changes were observed in the expression levels of the five *ERFs* in non-treated samples. By contrast, a significant increase was denoted in the accumulation of the five transcription factors in non-treated samples stored for 11 days at 0 °C. In RG red table grapes, the application of the gaseous treatment slightly induced the expression of *VviERF069*, with a significant increase observed for *VviERF6L7* (Figure 6).

In both cultivars, the accumulation of *VviERF069* correlated with changes in *VviERF6L7* (SS, r = 0.820, *p* < 0.01; RG, r = 0.757, *p* < 0.01). Moreover, changes in the *ERF2* gene expression showed a positive correlation with *ERF10* and *ERF11* in both cultivars (*ERF2*/*ERF10*; SS, r = 0.876, *p* < 0.01; RG, r = 0.624, *p* < 0.05; *ERF2*/*ERF11*; SS, r = 0.619, *p* < 0.05; RG, r = 0.730, *p* < 0.01) (Appendix A).

### 3.6. Effect of a 3-Day CO_2_ Treatment on Genes Codifying Cell Wall-Degrading Enzymes in the Rachis of SS and RG Table Grapes Stored at 0 °C

To discern whether the expression of *VviPG*, *VviEXP*, *VviXYL* and *VviPME*, which codified cell wall-degrading enzymes, could be related to rachis browning development, we determined their expression patterns in both CO_2_-treated and non-treated samples of red and white table grape bunches (Figure 7). The application of high levels of CO_2_ avoided the sharp increase observed in the accumulation of *VviPG* transcripts in non-treated samples of SS and RG bunches after 3 days at 0 °C. However, the accumulation of *VviPG* continued increasing in the non-treated rachis of SS after 11 days at 0 °C, while in the CO_2_-treated ones it resembled the levels recorded in freshly harvested samples. Nevertheless, an opposite pattern of *VviPG* accumulation was observed in RG bunches after 11 days at 0 °C. Thus, the levels of *VviPG* decreased in non-treated rachis to levels similar to those at time 0, whereas a sharp increase was denoted in CO_2_-treated samples. At the end of the storage period, no significant differences were observed in the *VviPG* gene expression between treated and non-treated samples, obtaining levels similar to those at time 0.

The pattern of *VviEXP1* and *VviXTH* gene expression was very similar in both cultivars. Although the storage at 0 °C decreased the accumulation of both transcripts, this decrease was significantly higher in those samples treated with high levels of CO_2_. It is important to note that in both cultivars the expression of *VviEXP1* and *VviXTH* was always lower than that observed in freshly harvested fruit. In this sense, a significant positive correlation (*p* < 0.01) was observed between the expression of both genes in both cultivars (Appendix A). Moreover, the application of high levels of CO_2_ also affected the expression of *VviPME*, decreasing the levels in comparison with non-treated samples. In this case, a transitory increase in *VviPME* accumulation was observed in non-treated samples on day 3, but, in general, the expression was always higher in non-treated samples until the end of the storage period, when similar levels were recorded in both treated and non-treated rachises.

## 4. Discussion

Progress in postharvest knowledge of table grapes has focused mainly on the prevention of gray mold infection caused by *B. cinerea*. However, less is known about rachis browning, which develops rapidly during the storage of table grapes at low temperatures. Rachis senescence influences the appearance of bunches and, although this only represents about 4% of their fresh weight [25], consumers associate a green rachis with freshness. Bunches with brown rachises tend to be classified as unattractive, which negatively affects their marketability. Rachis browning therefore represents a visual marker of freshness that plays an important role in consumer preference and food waste, so any advances in this area will be of great basic and applied impact.

We showed in a previous study that the application of a 3-day gaseous treatment at a low temperature improved the rachis browning of white (Superior Seedless) and red (Red Globe) table grape cultivars developed during storage at 0 °C for up to 28 days [9], but the mechanisms involved in this improvement were not analyzed. Thus, the present work delves into the study of the molecular mechanisms implicated in the beneficial effect of a 3-day gaseous treatment in maintaining rachis quality of both cultivars. The results showed that the development of rachis browning was linked to the non-enzymatic antioxidant system by the accumulation of phenolic compounds in both cultivars. Thus, rachis browning indexes and total phenolic content showed a positive correlation (SS, r = 0.807, *p* < 0.01; RG, r = 0.510, *p* < 0.05) (Appendix A). Moreover, a positive correlation was denoted between rachis browning and the antioxidant capacity determined through both methods, ABTS and FRAP, in both cultivars (SS, ABTS, r = 0.655, *p* < 0.01; FRAP, r = 0.745, *p* < 0.01) (RG, ABTS, r = 0.730, *p* < 0.01; FRAP, r = 0.936, *p* < 0.01) (Appendix A). This is one of the first works in which these analyses have been performed in relation to rachis browning since most studies analyzing the total phenolic content and antioxidant capacity during the postharvest of table grapes have been carried out on berries. Campos-Vargas et al. [26] showed an increase in the total phenolic content and the antioxidant capacity determined via the FRAP method in Red Globe rachises from bunches stored at 0 °C for up to 25 days under normal atmospheric conditions. However, this study did not mention the incidence of rachis browning in these bunches, so it is not possible to discern whether this was a response to cold stress caused by storage or was linked to the development of rachis browning.

Our results would appear to support that the application of a short-term high CO_2_ treatment at a low temperature has the effect of either maintaining constant levels or restricting the increase in the levels of total phenolic compounds in both cultivars. These results are in concordance with previous works, where the storage of fresh-cut Lotus roots under modified atmosphere packaging (MAP) with high concentrations of CO_2_ reduced the degree of browning and the content of total phenolic compounds [27]. Fruit with a higher phenolic content is therefore also known to present a higher potential for browning [28,29]. The polyphenol oxidase seems to play an important role since this enzyme oxidizes phenolic substrates to quinones that subsequently form brown, black and red pigments [17]. Thus, the application of a hexanal formulation to improve table grape quality suppressed PPO-related browning, thereby maintaining rachis freshness in Flame Seedless bunches [30]. In a previous work, we reported that the application of a 3-day gaseous treatment avoided the increase in the accumulation of *PPO* transcript observed in the non-treated rachis of Cardinal grapes at day 3 [3]. However, although we analyzed the *PPO* expression in CO_2_-treated and non-treated rachises of Red Globe and Superior Seedless table grapes, it was not possible to establish a relationship between the development of rachis browning or the phenol content (data not shown). Thus, the fact that storage under normal atmospheric conditions leads to a greater accumulation of phenolic compounds susceptible to oxidation could favor the development of browning in non-treated samples as more substrates are available, even though no differences in *PPO* expression were observed.

An important fact to note about this work is the role that *STS* gene expression seems to play as a molecular marker in the development of rachis browning. A positive correlation between the expression of all *STS* genes analyzed and the development of rachis browning was observed in both cultivars (Appendix A). Moreover, changes in *STS* gene expression were correlated significantly with the changes in the content of total phenolic compounds in both cultivars (Appendix A). Although these may be the first results in relation to the browning of rachis, Suehiro et al. [21] reported that skin browning of Shine Muscat grapes was associated with the expression of a *VvSTS* and the accumulation of *trans*-resveratrol. Adrian et al. [31] observed that *Botrytis cinerea* conidia treated with resveratrol possessed intracellular brown coloration, suggesting that discoloration resulted from the laccase-mediated oxidation of resveratrol. These authors suggested that these results imply that resveratrol could be a possible substrate for browning. On the other hand, it should not be forgotten that some phytoalexins such as stilbenoids present a broad spectrum of anti-pathogen activities, and their accumulation in grapes can promote a host defense response [32,33]. However, at this point, it is important to highlight that total decay was not recorded in Red Globe [9] and Superior Seedless [8] bunches after 28 days at 0 °C in any of the conditions assayed. Although more work is needed in this research area, this fact would indicate that the accumulation of *STS* transcripts in the rachis is more related to the development of browning than to a general defense response of the clusters against fungal attack.

Reactive oxygen species (ROS) metabolism is an important physiological metabolic activity in postharvest fresh crops related to fruit browning and senescence [34,35]. ROS levels in plant cells can be regulated using an ROS production-scavenging system which is related to the levels of antioxidant ingredients, such as total phenolic content, among others [36]. ROS accumulation may cause oxidative damage to lipids, forming toxic products such as MDA: a secondary end product of polyunsaturated fatty acid oxidation. Thus, an increase in lipid peroxidation and the concomitant production of MDA can cause further damage to cell membranes and accelerate the browning degree of fruit [37,38]. In this sense, we observed a positive correlation between rachis browning and MDA content in both cultivars (SS, r = 0.660, *p* < 0.01; RG, r = 0.890, *p* < 0.01). The fact that the application of high levels of CO_2_ at low temperatures decreases membrane lipid peroxidation suggests that the gaseous treatment may contribute to the ability of the rachis to cope with low temperature-induced oxidative stress. In this sense, the reduction in rachis browning of Superior Seedless clusters by the application of calcium nanoparticles coupled with oxalic acid has been linked to a reduction in MDA content [39]. Furthermore, our results reinforce the idea that the enzymatic antioxidant system, and in particular the catalase, plays a role in the control of rachis browning by the application of high levels of CO_2_ at low temperatures. Thus, a negative correlation between rachis browning and *CAT* expression has been established in both cultivars (SS, r = −0.481, *p* < 0.05; RG, r = −0.489, *p* < 0.05) (Appendix A). Overall, these results are in line with our previous work on red Cardinal table grapes [3], reinforcing the idea that one of the mechanisms involved in the reduction in rachis browning in CO_2_-treated bunches appears to be the activation of the enzymatic antioxidant system, controlling the oxidative stress that takes place during storage at low temperatures. Moreover, our results are in concordance with other postharvest treatments applied to reduce rachis browning. Thus, the reduction in rachis browning by the application of a coating treatment blended with salicylic acid was linked to the increase in antioxidant enzyme activities, such as APX and CAT, and the reduction in MDA content [40].

Although grapes are non-climacteric fruit, different lines of evidence point to the ethylene hormone as a major factor in rachis browning. Thus, 1-methylcyclopropane (1-MCP), a potent inhibitor of ethylene action, delayed rachis browning in three table grape varieties, whereas ethylene tended to enhance it [1]. The potential involvement of high CO_2_ in the inhibition of ethylene action is well known [41]. However, this response could depend on the specific tissue analyzed. The application of a short-term gaseous treatment induced *ACO* and an *ACS*-like gene expression in the skin and pulp of detached wine grapes [42] and Cardinal table grapes [3]. Nevertheless, the reduction in rachis browning by the application of high levels of CO_2_ has been linked to a decrease in *ACS1* gene expression in Cardinal table grapes [3]. Similarly, the present work showed a positive correlation between rachis browning and *ACS1* expression in Superior Seedless bunches (r = 0.516, *p* < 0.05), reinforcing the idea that ethylene could play a role in rachis browning, although it seems to be cultivar dependent. In this sense, Ye et al. [43] proposed that the expression of *VvACS1*, which was organ-specific, played an important role in rachis senescence.

ABA is another hormone to be considered in rachis browning due to its involvement in non-climacteric fruit senescence processes. However, studies conducted to date have yielded inconclusive results. The application of ABA to Shine Muscat grape clusters at the veraison stage significantly increased the severity of skin browning [10]. By contrast, treatment with ABA at veraison improved the rachis quality of Crimson Seedless table grapes during storage [11]. With reference to ABA biosynthetic genes, we could not establish a link between rachis browning and *NCED1* gene expression in CO_2_-treated and non-treated Cardinal table grapes [3]. However, this does not seem to be a common finding, since a positive correlation was observed in the present work between rachis browning development in Red Globe bunches and the pattern of expression of *VviNCED1* (r = 0.475; *p* < 0.05) and *VviNCED2* (r = 0.689; *p* < 0.01) (Appendix A). Moreover, it is important to note that ABA treatment leads to an increased accumulation of phenolic compounds in berries [10] and that, in our case, the total phenolic content of the rachis samples significantly correlated with the expression of ABA biosynthetic genes (SS; *VviNCED2*, r = 0.801, *p* < 0.01) (RG; *VviNCED1*, r = 0.654, *p* < 0.01; *VviNCED2*, r = −0.728, *p* < 0.01) (Appendix A). Therefore, the role of ABA in rachis browning could be through the accumulation of phenolic compounds; a fact that could be related to the higher browning index in the non-treated samples, since they are substrates susceptible to oxidation, giving rise to brown, black and red pigments.

*ERF* genes have been characterized in different plants, where they are involved in responses to biotic and abiotic stress [44,45]. In addition, ERFs are key targets for investigating transcriptional regulatory functions, not only in fruit development and ripening but also in senescence [46,47]. The effect of high levels of CO_2_, activating the expression of *ERFs*, has also been described in kiwifruit [48] and persimmon [49], but it is important to note that in both cases the effect of the gaseous treatments was assayed at 20 °C. The *ERF* family of transcription factors in table grapes seems to play a role in the beneficial effect of the gaseous treatment in Cardinal berry quality stored at low temperatures [13]. However, less is known about their involvement in the development of rachis browning, so it is interesting to know whether these transcription factors could be involved in the molecular control of rachis browning triggered by the application of high levels of CO_2_. Although our results showed differences according to the cultivar studied, they confirmed that the application of a 3-day gaseous treatment at 0 °C activated their accumulation, inducing the expression of *VviERF069* and *VviERF2* in the rachis of SS bunches and *VviERF6L7* in both cultivars. These results are in line with other works, where different technologies applied to delay the browning of fresh-cut lotus root, such as vacuum packaging and MAP followed by storage at 4 °C or low-temperature storage, alone induced the expression of different *ERFs* [27,50]. The fact that *VviERF6L7* presented a similar pattern of expression in the rachis of Cardinal bunches [13], Superior Seedless and Red Globe (present work) makes this gene a potential marker of the beneficial effect of the CO_2_ treatment in rachis samples.

Cell wall-related enzymes, such as PG, PME, XTH and EXP, play important roles in polysaccharide modification and enzymatic disassembly of the plant cell wall. Furthermore, postharvest fruit senescence has been generally correlated with the activity of several cell wall-degrading enzymes [51]. Recently, Li et al. [52] indicated the relationship between cell wall degradation induced by *Botrytis cinerea* and rachis browning. Moreover, the application of nano-calcium coupled with different concentrations of oxalic acid improved rachis browning, minimizing cell wall-degrading enzyme activities [39]. Notable variances exist in the response of Superior Seedless grapevine scion to the use of four rootstocks, in which the effect on the rachis browning incidence was examined. The ‘1103 Paulsen’ was the most effective in controlling Superior Seedless rachis browning by reducing the oxidation of the phenolic compounds and minimizing the activity of PG [53]. Our results reinforce the hypothesis that the control of wall-degrading enzymes would help to reduce the development of rachis browning. Although it was not possible to establish a statistical correlation between rachis browning and the genes analyzed, the application of the gaseous treatment generally reduced the expression levels of cell wall-degrading enzyme-related genes analyzed in both cultivars. PME and PG are two important enzymes that are responsible for breaking down the pectin component of the cell wall. PME results in demethylated pectins that can be degraded by PG. In this sense, it is important to highlight the effect of the application of high levels of CO_2_ on the reduction in *PG* and *PME* transcript levels, which showed a sharp increase in the non-treated rachis.

## 5. Conclusions

The application of a short-term high CO_2_ treatment at a low temperature on white and red table grapes activated specific responses in the rachis that may participate in the control of browning during postharvest. In both cultivars, both the non-enzymatic and enzymatic antioxidant systems seem to participate in the improvement of rachis browning by the application of high CO_2_ levels, either by preventing the induction that takes place in total phenolic content, antioxidant capacity and *VviSTSs* expression in non-treated rachis, or by inducing the expression of genes of the enzymatic antioxidant system. Moreover, this work reinforces previous results in Cardinal tables grapes [3], which indicated that ethylene could play a role in the control of rachis browning since the gaseous treatment avoided the activation of ethylene biosynthesis genes. However, the role of ABA appears to be cultivar dependent as a positive correlation between browning and ABA regulatory gene expression was only observed in Red Globe. Finally, the storage of table grapes under normal atmosphere conditions induced cell wall-degrading enzyme-related genes implicated in the hydrolysis of cell wall materials, resulting in decreased intercellular connections, which, coupled with the fact that these samples had a higher content of phenolic compounds susceptible to oxidation, could favor the development of browning.

## Figures and Tables

**Figure 1 ijms-23-13304-f001:**
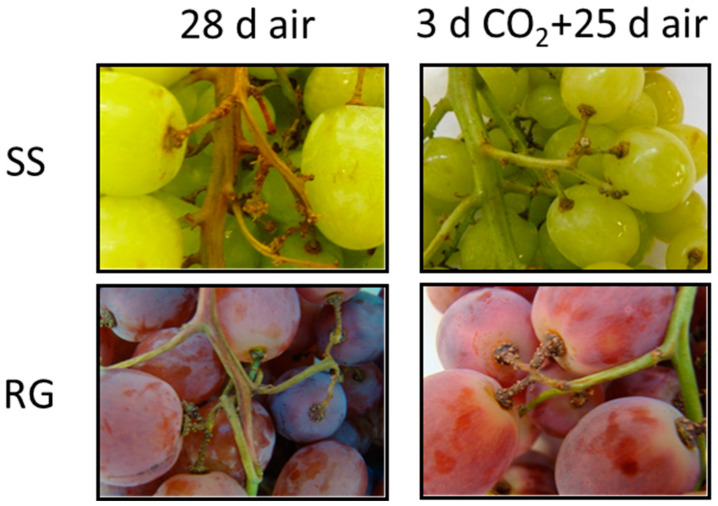
Effect of low temperature and a 3-day high CO_2_ treatment on rachis appearance of Superior Seedless (SS) and Red Globe (RG) table grapes stored for 28 days at 0 °C.

**Figure 2 ijms-23-13304-f002:**
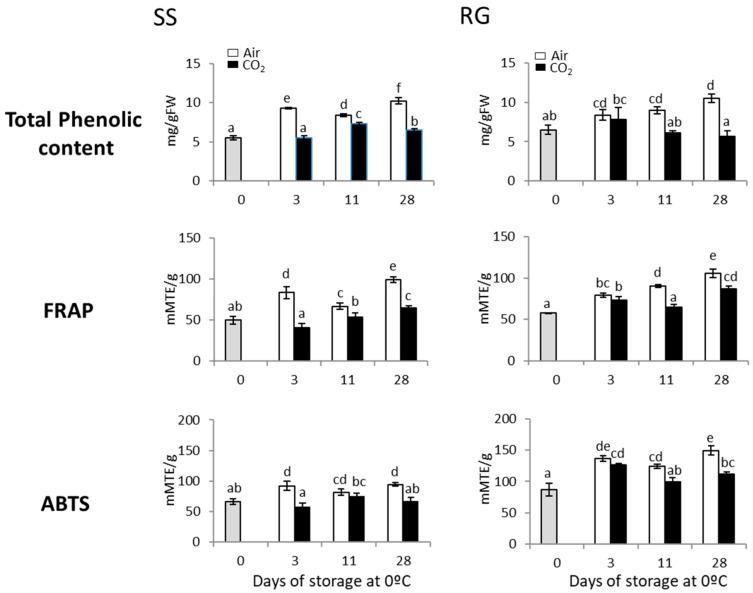
Changes in total phenolic content and antioxidant activity, determined via FRAP and ABTS, in the rachis of non-treated and CO_2_-treated Superior Seedless (SS) and Red Globe (RG) table grapes stored for 28 days at 0 °C. Values are the mean ± SD, *n* = 6. Different letters on the bars indicate that the values are statistically different using the Tukey-b test (*p* < 0.05).

**Figure 3 ijms-23-13304-f003:**
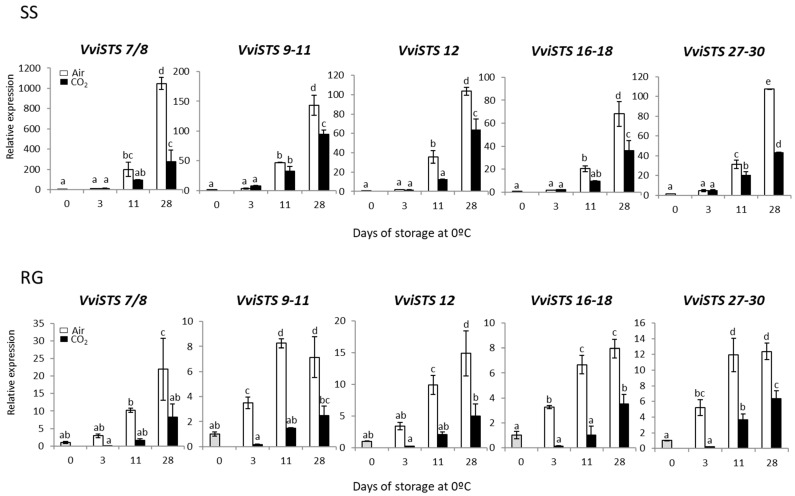
Effect of low temperature and a 3-day high CO_2_ treatment on the expression of different *VviSTSs* in the rachis of Superior Seedless (SS) and Red Globe (RG) table grapes stored for 28 days at 0 °C. Transcript levels were assessed through RT-qPCR and normalized using *Actin1* as reference gene. Results were calculated relative to a calibrator sample (time 0) using the formula 2^−ΔΔCt^. Values are the mean ± SD, *n* = 6. Different letters on bars indicate that the means are statistically different using the Tukey-b test (*p* < 0.05).

**Figure 4 ijms-23-13304-f004:**
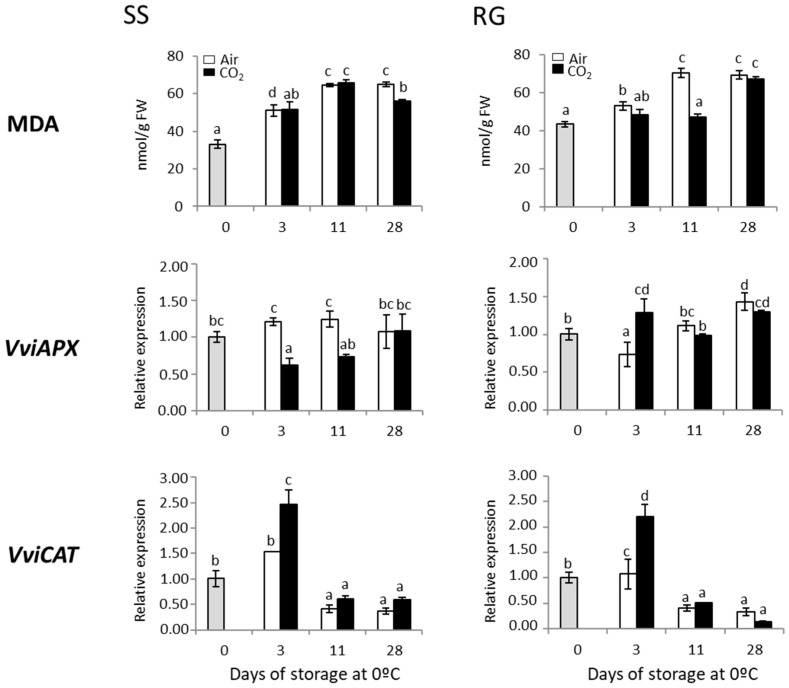
Effect of low temperature and a 3-day high CO_2_ treatment on the accumulation of MDA (nmol/g FW) and the expression of *VviAPX* and *VviCAT* in the rachis of Superior Seedless (SS) and Red Globe (RG) table grapes CO_2_-treated and non-treated and stored for 28 days at 0 °C. Transcript levels were assessed through RT-qPCR and normalized using *Actin1* as reference gene. Results were calculated relative to a calibrator sample (time 0) using the formula 2^−ΔΔCt^. Values are the mean ± SD, *n* = 6. Different letters on bars indicate that the means are statistically different using the Tukey-b test (*p* < 0.05).

**Figure 5 ijms-23-13304-f005:**
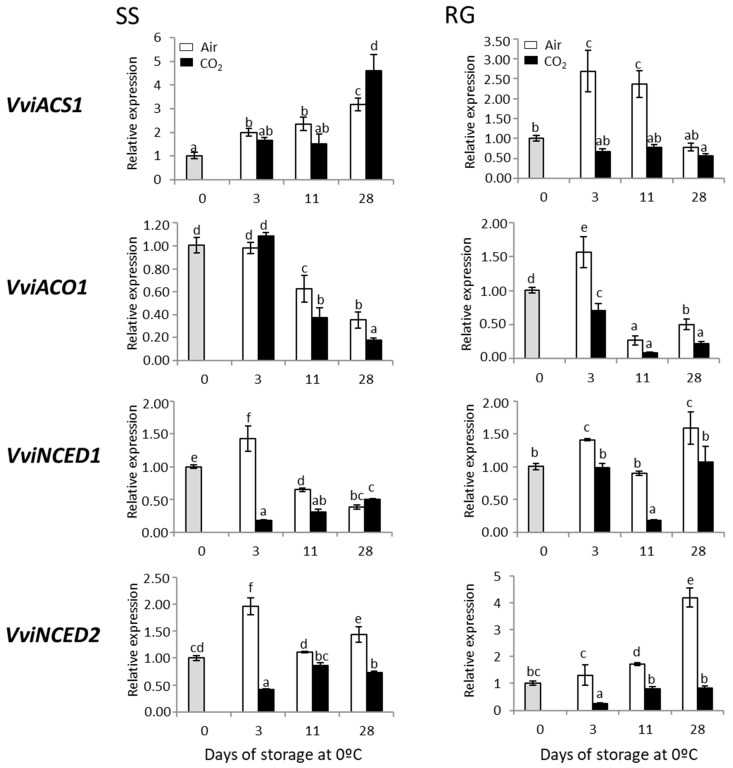
Effect of low temperature and a 3-day high CO_2_ treatment on the expression of ethylene (*VviACS1* and *VviACO1*) and ABA regulatory (*VviNCED1* and *VviNCED2*) genes in the rachis of Superior Seedless (SS) and Red Globe (RG) table grapes stored for 28 days at 0 °C. Transcript levels of each gene were assessed through RT-qPCR and normalized using *Actin1* as reference gene. Results were calculated relative to a calibrator sample (time 0) using the formula 2^−ΔΔCt^. Values are the mean ± SD, *n* = 6. Different letters on bars indicate that the means are statistically different using the Tukey-b test (*p* < 0.05).

**Figure 6 ijms-23-13304-f006:**
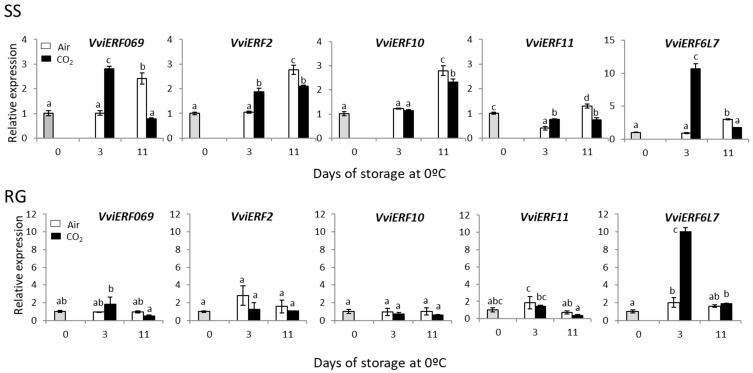
Effect of low temperature and a 3-day high CO_2_ treatment on the expression of different *VviERFs* transcription factors in the rachis of Superior Seedless (SS) table grapes stored for 28 days at 0 °C. Transcript levels of each gene were assessed via RT-qPCR and normalized using *Actin1* as reference gene. Results were calculated relative to a calibrator sample (time 0) using the formula 2^−ΔΔCt^. Values are the mean ± SD, *n* = 6. Different letters on bars indicate that the means are statistically different using the Tukey-b test (*p* < 0.05).

**Figure 7 ijms-23-13304-f007:**
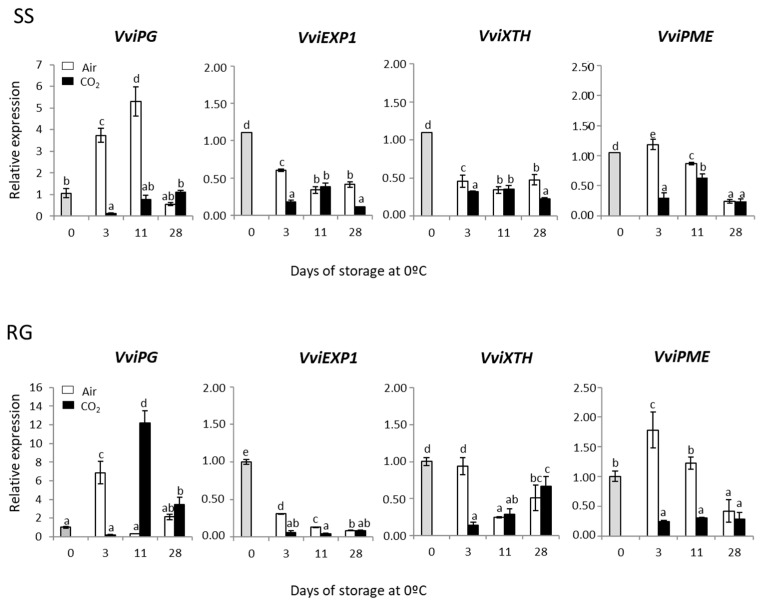
Effect of low temperature and a 3-day high CO_2_ treatment on the expression cell wall-related genes (*VviPG*, *VviEXP1*, *VviXTH* and *VviPME*) in the rachis of Superior Seedless (SS) and Red Globe (RG) table grapes stored for 28 days at 0 °C. Transcript levels were assessed via RT-qPCR and normalized using *Actin1* as reference gene. Results were calculated relative to a calibrator sample (time 0) using the formula 2^−ΔΔCt^. Values are the mean ± SD, *n* = 6. Different letters on bars indicate that the means are statistically different using the Tukey-b test (*p* < 0.05).

## Data Availability

Not applicable.

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
