# Peer review of "Short-Term Gaseous Treatments Improve Rachis Browning in Red and White Table Grapes Stored at Low Temperature: Molecular Mechanisms Underlying Its Beneficial Effect"

_ijms, 2022, doi:10.3390/ijms232113304_

Round 1

Reviewer 1 Report

The rachis quality is a key issue for table grape postharvest. Romero et al. present here an interesting set of data, showing a positive impact of a transient CO2 treatment on rachis quality. They also bring some new insights about molecular mechanisms that may be involved, in particular  cell wall-degrading enzymes-related genes and hormone status.

This will be a valuable contribution to this scientific field.

Author Response

We thank reviewer for his/her kindly comments.

Reviewer 2 Report

Major comments,

1. Authors should reveal how the CO2 treatment regulate the fruit quality in grape. Especially, how did author confirm CO2 treatment regulate ehtylene production or ABA content? Why choose antioxidant capacity and phenolic content as the object of study?

2. What reason does authors select the ACS1, ACO1, NCED1 and NCED2 to explore the CO2 effect mechanism?

3. There was no data of fruit firmness changement, so how to conclude that cell wall-degrading enzymes affect the fruit quality of grape? And the enzymes activities should be added in this manuscript.

4. There are some problems in the design of the manuscript. From the result, we could not get the main idea, is the author want to elucidate the rachis browning? But they didnt show the browing index or some related physiological indexes changement. And they also didnt show the direct relationship of the browning and the ethylene biosynthesis and ABA synthesis.

Author Response

We appreciate your comments that have helped us to improve the manuscript.

  1. Authors should reveal how the CO2 treatment regulate the fruit quality in grape.

Our previous works already revealed the effect of the short-term high CO2 levels in maintaining the quality of table grapes from different cultivars. Please see references 6, 7, 8, 9, 13 and 16 of the present manuscript. In these works, we have showed that the application of a 3-day gaseous treatment controlled total decay and water loss and we have explored the molecular mechanisms implicated in this beneficial effect. Since the molecular mechanisms involved in the maintenance of rachis quality by the application of high levels of CO2 have been little studied, we propose this study with the aim of deepening our knowledge in this issue in two different table grapes cultivars, Superior Seedless and Red Globe.  

In the introduction of the first submitted version we indicated that “The objective of the present work is to determine the role of the non-enzymatic (total phenolic content, antioxidant capacity, expression of VviSTSs) and enzymatic antioxidant (expression of catalase (VviCAT) and ascorbate peroxidase (VviAPX)) systems in the development of rachis browning in two table grapes cultivars. We also want to elucidate the correlation between the maintenance of rachis quality and the modulation of key ethylene (ACS1 and ACO1) and ABA (VviNCED1 and VviNCED2) regulatory genes, as well as different ERFs genes. Finally, we analyzed whether the differences denoted in rachis senescence between CO2-treated and non-treated bunches were related to the expression of genes (polygalaturonase (VviPG), expansin (VviEXP), xylanase (VviXYL), pectin methylesterase (VviPME) and cellulase (VviCEL), which encode different cell wall-degrading enzymes.

For this reason, we have not included results on berries, as we have published works on this subject and in the present work we have focused only on the molecular mechanisms involved in the improvement of rachis quality by the application of high levels of CO2.

Especially, how did author confirm CO2 treatment regulate ehtylene production or ABA content?

At this point, we would like to make a clarification, since throughout the manuscript we did not confirm that the application of high levels of CO2 regulated ethylene production and ABA content. Different lines of evidence point to the ethylene and ABA as hormones affecting rachis browning. As we indicated in the discussion section, “The potential involvement of high CO2 in the inhibition of ethylene action is well known [41]”. In this sense, we speculated about the effect of the gaseous treatment controlling rachis browning by regulating the expression of key ethylene and ABA regulatory genes.

In relation to the ethylene biosynthetic genes, we argued that our results reinforced previous results obtained in Cardinal table grapes where the reduction in rachis browning by the application of high levels of CO2 was linked to a decrease in ACS gene expression [3]. Thus, in the present work we observed a positive correlation between rachis browning and ACS1 expression in Superior Seedless bunches (r=0.516, p<0.05), reinforcing the idea that the ethylene could play a role in rachis browning. However, the fact that these results were not obtained in Red Globe bunches highlight that seems not to be a common effect of gaseous treatment, but is rather cultivar dependent.

The analysis of the ABA-related genes (VviNCED1 and VviNCED2) revealed that in both cultivars, their expression levels were higher in non-treated rachis compared to CO2-treated ones. However, a positive correlation between browning and ABA regulatory gene expression was only denoted in Red Globe. As we indicated in the discussion section, “it is important to note that ABA treatment leads to an increased accumulation of phenolic compounds in berries [10] and that in our case, the total phenolic content of the rachis samples significantly correlated with the expression of ABA biosynthetic genes”. In order to clarify this issue, we have added  to the discussion section “Therefore, the role of ABA in rachis browning could be through the accumulation of phenolic compounds, a fact that could be related to the higher browning index in the non-treated samples, since they are substrates susceptible to oxidation, giving rise to brown, black and red pigments” (Lines 577-581).

Why choose antioxidant capacity and phenolic content as the object of study?

It is known that the phenolic compounds play an important role in fruit visual appearance and affected the non-enzymatic antioxidant capacity. However, it has not been studied so far whether the application of a short-term high CO2 levels treatment can affect phenolic content and antioxidant capacity in the rachis. .

In order to clarify this issue we have improved the introduction section and we have added. “Previous works suggested that the phenylpropanoid pathway, which is one of the best-known metabolic pathways in plant cells, appears to be involved in rachis browning [3,16]. Phenolic compounds play an important role in fruit visual appearance. Moreover, monohydroxyphenols and orthodihydroxyphenols are substrates of plant polyphenol oxidases (PPO) enzymes, that produce brown polymers, affecting fruit quality [17]. However, so far it is not known whether the application of high levels of CO2 could affect the phenol content in the rachis, which play an important role in the non-enzymatic antioxidant capacity” (Lines 123-130)

  1. What reason does authors select the ACS1, ACO1, NCED1 and NCED2 to explore the CO2 effect mechanism?

We have selected these genes based on previous results obtained in Cardinal grape rachis. In accordance with our results, Ye et al [44] proposed that the expression of VvACS1, was related to rachis senescence. Thus, we have added a sentence in the discussion section including this reference.

“In this sense, Ye et al [44] proposed that the expression of VvACS1, which was organ-specific, played important roles in rachis senescence” (Lines 559-561)

  1. There was no data of fruit firmness changement, so how to conclude that cell wall-degrading enzymes affect the fruit quality of grape? And the enzymes activities should be added in this manuscript.

As indicated above, the present work focuses on analyzing the molecular mechanisms involved in rachis browning and results from berries were not included. Therefore, it is not concluded that cell wall degrading enzymes affect grape fruit quality.

In relation to this issue, as we have indicated in the introduction section, the main objective of this work was to analyze whether the differences denoted in rachis senescence between CO2-treated and non-treated bunches were related to the expression of genes (polygalaturonase (VviPG), expansin (VviEXP), xylanase (VviXYL), pectin methylesterase (VviPME) and cellulase (VviCEL), which encode different cell wall-degrading enzymes.

We appreciate the reviewer's suggestion; however, we have focused this work on characterizing the molecular mechanisms through gene expression studies, so enzymatic activities were not analyzed.

  1. There are some problems in the design of the manuscript. From the result, we could not get the main idea, is the author want to elucidate the rachis browning? But they didn’t show the browing index or some related physiological indexes changement. And they also didn’t show the direct relationship of the browning and the ethylene biosynthesis and ABA synthesis.

All the results showed in the present work were related to studies performed to elucidate the mechanism implicated in rachis browning. Thus, the title, objectives, material and methods, results, figure captions and discussion refer to this tissue; how storage at low temperature induce browning, the role of gaseous treatment controlling it and the molecular mechanisms involved.

To clarify this issue we have included a sentence in the introduction, before enumerating the different objectives of the work:

“The present work proposes a study in order to elucidate the molecular mechanisms implicated in the maintenance of rachis quality by the application of short-term gaseous treatments in an early-ripening, white table grape cultivar (Superior Seedless, SS) and a mid-ripening, red cultivar (Red Globe, RG)” (Lines 142-145)

We decided not to show the rachis-browning index to avoid repeating previous published results. As we have indicated in the first version of this manuscript, these results were already published in a previous paper studying the effect of the gaseous treatment in four different table grapes cultivars in maintaining cluster quality modulating the content of stilbenoids and regulating the expression of STSs (reference 9). However, to improve this issue we have incorporate:

“The application of a 3-day treatment with high levels of CO2 was effective in controlling rachis browning in RG and SS bunches stored for up to 28 days at 0 ºC [rachis browning index (RG, 28 d, Air, 3.66 ± 0.47 b; CO2, 2.83 ± 0.23 a), (SS, 28 d, Air 3.00 ± 0.40 b; CO2, 2.00 ± 0.35 a) [9]; Figure 1)” (Lines 227-229).

In reference to the ethylene (ACS1 and ACO1) and ABA (VviNCED1 and VviNCED2) regulatory genes, we speculate about the putative role of their expression as markers of browning. Moreover, our results reinforce previous results obtained by our research group, as well as others, as we have indicated along the manuscript. 

Round 2

Reviewer 2 Report

no other suggestions.